# Heme Oxygenase-1 Inhibition Modulates Autophagy and Augments Arsenic Trioxide Cytotoxicity in Pancreatic Cancer Cells

**DOI:** 10.3390/biomedicines11092580

**Published:** 2023-09-20

**Authors:** Iman M. Ahmad, Alicia J. Dafferner, Ramia J. Salloom, Maher Y. Abdalla

**Affiliations:** 1Department of Clinical, Diagnostic, and Therapeutic Sciences, University of Nebraska Medical Center, Omaha, NE 68198, USA; iman.ahmad@unmc.edu; 2Department of Pathology and Microbiology, University of Nebraska Medical Center, Omaha, NE 68198, USA; adafferner@unmc.edu (A.J.D.); ramia.salloom@unmc.edu (R.J.S.)

**Keywords:** arsenic trioxide (ATO), Heme Oxygenase-1, autophagy

## Abstract

Pancreatic ductal adenocarcinoma (PDAC) is the most prevalent form, accounting for more than 90% of all pancreatic malignancies. In a previous study, we found that hypoxia and chemotherapy induced expression of Heme Oxygenase-1 (HO-1) in PDAC cells and tissues. Arsenic trioxide (ATO) is the first-line chemotherapeutic drug for acute promyelocytic leukemia (APL). ATO increases the generation of reactive oxidative species (ROS) and induces apoptosis in treated cells. The clinical use of ATO for solid tumors is limited due to severe systemic toxicity. In order to reduce cytotoxic side effects and resistance and improve efficacy, it has become increasingly common to use combination therapies to treat cancers. In this study, we used ATO-sensitive and less sensitive PDAC cell lines to test the effect of combining HO-1 inhibitors (SnPP and ZnPP) with ATO on HO-1 expression, cell survival, and other parameters. Our results show that ATO significantly induced the expression of HO-1 in different PDAC cells through the p38 MAPK signaling pathway. ROS production was confirmed using the oxygen-sensitive probes DCFH and DHE, N-acetyl cysteine (NAC), an ROS scavenger, and oxidized glutathione levels (GSSG). Both ATO and HO-1 inhibitors reduced PDAC cell survival. In combined treatment, inhibiting HO-1 significantly increased ATO cytotoxicity, disrupted the GSH cycle, and induced apoptosis as measured using flow cytometry. ATO and HO-1 inhibition modulated autophagy as shown by increased expression of autophagy markers ATG5, p62, and LC3B in PDAC cells. This increase was attenuated by NAC treatment, indicating that autophagy modulation was through an ROS-dependent mechanism. In conclusion, our work explored new strategies that could lead to the development of less toxic and more effective therapies against PDAC by combining increased cellular stress and targeting autophagy.

## 1. Introduction

Pancreatic ductal adenocarcinoma (PDAC) is an aggressive cancer with a high mortality rate [1]. According to the American Cancer Society, PDAC is the third leading cause of cancer death in men and women combined (about 3% of all cancers in the USA) with an estimated 50,550 people that will die of PDAC in 2023 [2].

One of the major problems in PDAC treatment is drug resistance, thus making new drug strategies ineffective in clinical trials [3]. Understanding the basis of drug resistance will lead to development of novel therapeutic strategies, including combinations of two or more drugs. This will increase the efficacy of current treatments used against PDAC.

Arsenic trioxide (As_2_O_3_) (ATO) was approved by the US Food and Drug Administration (FDA) as the first-line treatment of APL [4]. An increasing body of evidence has shown ATO has demonstrated high efficiency in treating multiple solid tumors [5,6]. Increased survival and less extrahepatic metastasis were reported in hepatocellular carcinoma (HCC) patients who received ATO alone or in combination with other drugs [7]. Studies showed that ATO induced apoptosis and growth inhibition in pancreatic cancer cells [8,9]. In advanced PDAC patients, ATO was given after gemcitabine therapy in a phase II clinical trial, but the results were not encouraging [8]. It is very clear that novel combinations with ATO can lead to potent therapeutic drugs for drug-resistant PDAC patients. It is believed that side effects and drug resistance due to high dose requirement partly account for the failure of ATO treatment in solid tumors [10]. Therefore, new strategies are needed to reduce the toxicity of and eliminate resistance to ATO [10].

Heme oxygenase catalyzes the degradation of heme into biliverdin, carbon monoxide (CO), and free iron [11]. Among the three isoforms (HO-1, HO-2, and HO-3), HO-1 is the inducible isoform [11]. HO-1 regulates cell proliferation and accelerates PDAC angiogenesis [12]. Expression of HO-1 is induced in cancer cells by different stimuli including radiation and chemotherapy [13,14]. Poorer survival was associated with increased expression of HO-1 in different cancer patients including PDAC [15,16]. It was shown that anti-apoptotic activities of HO-1 protect cancer cells against apoptosis [15,17]. Moreover, studies have shown that upregulation of HO-1 is correlated with modulation of autophagy [18,19]. Metalloporphyrin compounds (MPs) including Tin protoporphyrin IX (SnPP) and Zinc protoporphyrin IX (ZnPP) bind HO with higher affinity than heme [14,15]. The inhibition of HO by MPs has been reported in animal and human subjects. Our lab and others have demonstrated that using HO-1 inhibitors sensitizes tumor cells for chemotherapy [14]

Previously, our lab and others have shown that tumor hypoxia and chemotherapy (including Gemcitabine and nab-Paclitaxel) induce HO-1 in PDAC cells and in tumors obtained from PDAC patients. Inhibiting HO-1 sensitized PDAC cells and tumors to chemotherapy [14,15]. In this work, we will study the combined use of HO-1 inhibitors and ATO against PDAC cells and find out the role of HO-1 in modulating autophagy. ATO was reported to play a potent role in perturbing autophagy in cancer cells [20]. Multiple proteins are involved in the various stages of autophagy, and studying the mechanisms underlying this process will help develop new treatment protocols in PDAC. 

In this work, our results showed that ATO and HO-1 inhibition increased ROS and apoptosis, disrupted the GSH, system and modulated autophagy through ROS-mediated mechanisms.

## 2. Materials and Methods

### 2.1. Western Blot

PDAC cell lines (MiaPaca-2, CD18 HPAF, Capan-1, T3M4, and Colo357) were either obtained from the ATCC or a kind gift from Dr. Tony Hollingsworth’s lab at the University of Nebraska Medical Center (UNMC). Cells were cultured in DMEM with 7% heat-inactivated fetal bovine serum. Cells were placed in a 5% CO_2_ incubator at 37 °C. Cells were treated with arsenic trioxide (LabChem, Zelienople, PA, USA) at varying concentrations in NaOH with normal saline. Tin protoporphyrin IX dichloride (SnPP) and Zinc protoporphyrin (ZnPP) were obtained from Santa Cruz Biotechnology (Dallas, TX, USA). The treated cells were incubated for 24 h at 37 °C with 5% CO_2_ (normoxia samples) or 5% CO_2_ and 2% O_2_ (hypoxia samples). Protein was collected using freshly made lysis buffer (RIPA buffer with 1% protease inhibitor and 1% 0.5 M EDTA) and sonication. Protein was quantified using the Bio-Rad DC Protein Assay Kit according to manufacturer instructions and measured using a BioTek Synergy H1 plate reader. Samples were run in 12% SDS gels, transferred to Immobilon-P PVDF membranes, and blocked with 5% milk. The primary antibody (HO-1 Enzo, NY; p38 Cell Signaling Technology (CST), MA, P-p38 CST, MA, or Beta Actin CST, MA) was diluted 1:1000 in 5% milk. The secondary antibody (goat anti-rabbit IgG Enzo, NY, or anti-mouse IgG CST, MA) was diluted 1:5000 in 5% milk and incubated at room temperature for 1 h on a shaker. Blots were developed using Azure Biosystems Radiance Plus and imaged using Azure c600. The band intensity of protein expression in Western blots was measured with ImageJ 1.54d software and was normalized to β-actin. 

### 2.2. Flow Cytometry for Apoptosis

PDAC cells were treated with Arsenic Trioxide at indicated time points. Samples were tested for apoptosis using an eBioscience Annexin V Apoptosis Detection Kit according to manufacturer instructions (ThermoFisher, Waltham, MA, USA).

### 2.3. Non-Radioactive Cell Proliferation Assay (MTT)

An MTT assay was performed using Promega’s MTT assay kit (Promega, WI, USA) according to the manufacturer’s protocol. Assays were read at 570 nm using a BioTek Synergy H1 plate reader.

### 2.4. Confocal Microscopy

PDAC cells were plated out in 4-well slide chambers (BD Falcon, NJ, USA) and treated with ATO Tin and/or Zinc Protoporphyrin (Santa Cruz Biotechnology, TX, USA). Cells were fixed with 4% paraformaldehyde and permeabilized using 0.1% Triton X-100. Non-specific binding was blocked by adding 1% goat serum. Anti-HO-1 primary antibody (Enzo, NY, USA) was used in 1% goat serum and incubated overnight at 4 °C in a humidified chamber. For the secondary antibody, we used Alexa Fluor 488 goat anti-rabbit IgG (H + L) (ThermoFisher, Waltham, MA, USA). Slides were mounted with Vectashield DAPI mounting media (Vector Laboratories, Newark, CA, USA). Images were obtained using a Zeiss LSM 710 at the UNMC core facility.

### 2.5. ROS Detection

For ROS experiments, the superoxide indicator dihydroethidium (DHE, ThermoFisher, Waltham, MA, USA) and the ROS indicator Dichloro-dihydro-fluorescein diacetate (CM-H2DCFDA, ThermoFisher, Waltham, MA, USA) reagents were added for 30 min at 37 °C on live PDAC cells cultured in slide chambers. Slides were analyzed using the Zeiss LSM 710. 

### 2.6. Lysosomal Staining

MiaPaca-2 cells were seeded in chamber slides (ThermoFisher, Waltham, MA, USA) and treated with different compounds for 24 h. The next day, cells were washed and stained with LysoTracker Red DND-99 probe (Invitrogen, Carlsbad, CA, USA) in triplicate wells. LysoTracker Red was diluted to a 60 nM concentration and applied for 1 h on live cells. Slides were then mounted with DAPI containing mounting media (Vectashield #H-1500). Slides were visualized using the Zeiss LSM 710 at the UNMC core facility. 

### 2.7. Transfection

PDAC cells were transfected with an HMOX1 Human shRNA Plasmid Kit (Origene, MD, USA) and Lipofectamine 3000 (ThermoFisher, Waltham, MA, USA). Transfection was performed according to the Lipofectamine 3000 protocol using 1 µg of plasmid DNA for each reaction. Selection medium containing puromycin was used to select for positive cells. Scramble plasmids were used as a control.

### 2.8. Autophagy

For confocal studies, PDAC cells were cultured and treated with ATO, HO-1 inhibitors, Zinc Protoporphyrin (Santa Cruz, TX, USA), Tin Protoporphyrin (Frontier Scientific, UT, USA), 1 mM N-acetyl cysteine (Sigma-Aldrich, MO, USA), or 25 µM Chloroquine (Cayman Chemical, MI, USA). Chamber slides were processed for confocal microscopy using primary antibody ATG5 (CST, MA, USA) at 1:500 and secondary antibody Alexa Fluor 488 goat anti-rabbit IgG (H + L) (ThermoFisher, Waltham, MA, USA) at 1:200, both in 1% goat serum. Western blots were run as described above. Primary antibodies were HO-1 (Enzo, NY, USA), ATG5 (CST, MA, USA), and LC3B (CST, MA, USA), along with secondary antibodies goat anti-rabbit IgG (Enzo, NY, USA) or anti-mouse IgG (CST, MA, USA).

### 2.9. Glutathione (GSH) Analysis

GSH and GSSG were measured using a GSH Quantification kit (DOJINDO Inc., Rockville, MD, USA) according to the manufacturer’s guidelines. In brief, PDAC cultures were washed with phosphate-buffered saline (PBS) and detached using a cell scraper. Cells were collected through centrifugation at 4 °C and 200× *g*, then resuspended in 10 mM hydrochloric acid and vortexed. The cells were lysed by two freeze/thaw cycles. Protein was quantified using a DC Protein Assay (Bio-Rad, Hercules, CA, USA). 

### 2.10. Statistical Analysis

All data are representative of at least 3 independent experiments and presented as the mean ± SEM. Statistical differences between three or more groups were calculated through one-way analysis of variance (ANOVA) with Bonferroni post hoc tests. All *p* values less than 0.05 were considered as statistically significant. Analyses were performed with GraphPad Prism version 9.4 for Windows (GraphPad Software, San Diego, CA, USA).

## 3. Results

### 3.1. ATO Treatment Induces HO-1 Expression in PDAC Cells

To examine the effects of ATO on HO-1 expression levels in PDAC cells, we treated different PDAC cell lines with a range of ATO concentrations and examined HO-1 using Western blot analysis and confocal microscopy. ATO increased HO-1 expression in PDAC cell lines (MiaPaca-2 and Capan-1), as shown by immunoblotting (Figure 1A). Similar induction was seen in another PDAC cell line, Colo-357. This significant increase in HO-1 expression was confirmed through confocal microscopy in Colo-357 cells, as shown in Figure 1B (*p* < 0.05). All three PDAC cell lines (Colo-357, MiaPaca-2, and Capan-1) showed a dose-dependent increase in HO-1 upon exposure to different concentrations of ATO.

### 3.2. ATO Induces Phosphorylation of p38 MAPK Pathway in Human PDAC Cells

Mitogen-activated protein kinases (MAPKs) such as p38 are known to activate and increase expression of HO-1 through an NF-E2-related factor 2 (Nrf2) pathway [14,21]. This led us to speculate whether ATO activates this pathway. Interestingly, ATO dose-dependently increased the phosphorylation of p38 MAPK. Addition of ATO to MiaPaca-2 and Capan-1 cells induced p38 MAPK phosphorylation (Figure 1C) at similar concentrations to those used in other experiments.

### 3.3. HO-1 Inhibition Enhanced ATO-Mediated Cell Killing in PDAC Cells

To explore the effects of HO-1 inhibition on PDAC response to ATO treatment, we used the MTT cell viability assay to analyze cell survival. ATO caused dose- and time-dependent cell death in multiple PDAC cell lines. Previously, it was shown that some PDAC cell lines are sensitive to ATO treatment and others are less sensitive [22]. Therefore, we decided to test HO-1 inhibition in both sensitive and less sensitive cells (Figure 2). Dose- and time-dependent effects were seen with ATO treatment. In ATO-sensitive cells, MaiPaca-2 cells, a dose-dependent effect was seen when cells were cultured with increased concentrations of ATO for 24 h (Figure 2A). Combining HO-1 inhibition with ATO reduced cell survival to 48% after 24 h and to 23% after 48 h culture as compared to ATO alone (Figure 2B). A similar but lesser effect was seen in less ATO-sensitive cells, Capan-1 (Figure 2C). Inhibiting HO-1 in combination with ATO treatment reduced the survival of Capan-1 to 63% and to 54% after 24 h and 48 h, respectively (Figure 2D). In T3M4 PDAC cells, combining ATO with HO-1 inhibition reduced proliferation to 41% after 24 h (Figure 2E). To confirm the pharmacological inhibition of HO-1, we created HO-1 gene knockdown (KD) in MiaPaca-2 cells. To test the HO-1 expression, cells were treated with HO-1 inducer Cobalt protoporphyrin (CoPP). The induction of HO-1 was significantly higher in parent cells (controls) as compared to HO-1 KD cells (Figure 2F). HO-1 KD cells showed a more than 70% reduction in survival as compared to the control when treated with ATO. This result indicates the crucial role of HO-1 induction in protecting PDAC cells from ATO effects and the promising result of combined treatment (Figure 2F).

### 3.4. HO-1 Inhibition Enhanced ATO Induced Apoptosis in PDAC Cells

Since cell survival was reduced by ATO and HO-1 inhibition, we hypothesized that apoptosis might be involved in this process. Apoptotic cells were detected using Annexin V and PI staining with flow cytometry. We aimed to analyze the combined effect of ATO and HO-1 inhibitors Tin protoporphyrin (SnPP) and Zinc protoporphyrin (ZnPP). Flow cytometry showed that ATO or HO-1 inhibition treatment increased apoptosis in PDAC cells. This increase was enhanced in the combined treatment (Figure 3A–C). In T3M4, the number of apoptotic cells was increased 2.9-fold in the ATO-treated group at 1 μM and 3.5-fold at 3 μM ATO as compared to the control group. Inhibiting HO-1 significantly increased the number of apoptotic cells 3.6- and 5.4-fold with 1 and 3 μM ATO treatment, respectively (*p* < 0.05). Similarly, in MiaPaca-2, inhibiting HO-1 using ZnPP increased ATO-induced apoptosis 2.9-fold (Figure 3C). No autofluorescence was detected in any of these cells (Appendix A). These results suggest that HO-1 inhibition promotes ATO-induced apoptosis in different PDAC cells.

### 3.5. ATO/HO-1 Inhibition Combination Stimulates ROS Generation and Disrupts ROS Scavenging in PDAC Cells

Previously, ROS production and oxidative-stress-induced apoptosis in tumor cells was reported as one mechanism of ATO action [23,24]. Therefore, we sought to measure ROS production and the major antioxidant compound glutathione (GSH). Direct measurement of ROS levels in MiaPaca-2 showed that both HO-1 inhibition and ATO increased ROS levels at 24 h, as measured by increased fluorescence in the ROS-sensitive probes DCFH and DHE (Figure 4A–C). A significant increase (4.5-fold) in DCF fluorescence was seen when cells were treated with ATO at 5 μM as compared to non-treated controls (*p* < 0.05). The fluorescence increased to 12.5-fold when cells were treated with 10 μM ATO for 24 h (*p* < 0.05). Both HO-1 inhibitors (SnPP and ZnPP) increased DCF fluorescence 3.5–4-fold. To confirm that increased fluorescence is due to increased ROS production, we treated the cells with the ROS scavenger NAC at 1 mM. NAC reduced DCF fluorescence to near control levels when added to ATO (Figure 4). Similarly, ATO increased DHE fluorescence in MiaPaca-2 cells 3.5-fold (Figure 4C).

The oxidative status of ATO- and/or HO-1 inhibitor-treated cells was evaluated by measuring the levels of oxidized (GSSG) and reduced (GSH) glutathione as GSH is a major ROS scavenger in cells. Combined treatments at 24 h caused augmented amounts of ROS in the two PDAC cell lines MiaPaca-2 and Capan-1, as reflected the disrupted GSH levels. A significant increase in GSSG was detected when HO-1 inhibition was combined with ATO (Figure 4D). To demonstrate whether induced ROS is part of the mechanism of action, we compared the survival effects in cells treated with ATO in the presence of the ROS scavenger NAC. NAC restored the suppressed proliferation of Colo-357, CD18/HPAF, and MiaPaca-2 cells (Figure 4E).

### 3.6. ATO and HO-1 Inhibitors Increased Lysosomal Staining in PDAC Cells

A crucial step in autophagy is the autophagosome acquiring degradative enzymes by fusing with the lysosome. Therefore, we sought to study the levels of lysosomes as a part of the pathways that lead to autophagy formation and degradation. LysoTracker-based fluorescent staining specific for lysosomal structures was used in our study to analyze the effects of ATO and HO-1 inhibition on lysosomes. A 3.4-fold increase in LysoTracker fluorescence intensity was observed for cells treated with ATO (5 µM). Both HO-1 inhibitors (ZnPP and SnPP) increased the fluorescence 2-fold. Combining HO-1 inhibitors with ATO significantly increased the fluorescence 4.3-fold (*p* < 0.05). Chloroquine (CQN) was used as the positive control (Figure 5A,B).

### 3.7. HO-1 Inhibition and ATO Regulate Autophagy through ROS-Mediated Mechanism

Our data show that ATO induces HO-1 and inhibition of HO-1 increases ROS and apoptosis and disrupts lysosomal accumulation in PDAC cells. We sought to explore the effect of ATO or HO-1 inhibition on autophagy flux. We evaluated the levels of three autophagy-specific markers, LC3B, p62, and ATG5. Data from confocal studies and Western blot analysis showed that inhibiting MiaPaca-2 cell proliferation with HO-1 inhibitors or ATO was accompanied by the formation of autophagosomes in the cytoplasm (Figure 6A), as demonstrated by significantly increased ATG5 staining in the cytoplasm (Green Fluorescence) (*p* < 0.05). A clear increase in autophagy-related proteins LC3B and ATG5 was observed in immunoblotting (Figure 6C). As a control, CQN, a known inhibitor of lysosomal degradation, was used to analyze the levels of LC3B and ATG5 in treated cells [25,26].

Our results showed that incubation of cells with CQN induced increases in ATG5 and LC3B levels as compared to the control. In the confocal study, ATO significantly increased ATG5 levels in MiaPaca-2 cells (*p* < 0.05). In addition, both HO-1 inhibitors significantly increased ATG5 (*p* < 0.05) (Figure 6A). Treating cells with the ROS scavenger NAC reduced ATG5 levels to near control levels (Figure 6A,B). This interesting observation indicates that increased expression of the autophagy-related protein ATG5 is an ROS-mediated process. 

To confirm our results, we treated PDAC cells with ATO and HO-1 inhibitors to test autophagy protein expression through Western blotting. ATO elevated ATG5 and LC3B in MiaPaCa-2 cells (Figure 6C). Interestingly, inhibiting HO-1 increased both autophagy markers in Mia-Paca-2 (Figure 6C). Again, we found that NAC could notably reverse the autophagy disruption by ATO or HO-1 inhibition. NAC restored cell viability by ameliorating ROS-mediated cell death. These results indicate that HO-1 inhibition interferes with and inhibits autophagic flux in PDAC cells. 

Additionally, p62 levels were higher in HO-1 inhibitor-treated cells, suggesting that inhibiting HO-1 inhibits autophagic flux (Figure 6C). The p62 protein, also called sequestosome 1 (SQSTM1), is a ubiquitin-binding scaffold protein that colocalizes with ubiquitinated protein aggregates. P62 accumulates when autophagy is inhibited, and decreased levels can be observed when autophagy is induced, so p62 may be used as a marker to study autophagic flux. The addition of NAC reduced autophagy marker expression (LC3), confirming that increased autophagy markers is ROS-mediated (Appendix A).

Collectively, our results suggest that HO-1 inhibition suppresses autophagy by interacting directly with autophagy proteins and disrupting the autophagosome network.

## 4. Discussion

Pancreatic ductal adenocarcinoma is one of the lethal malignant tumors, with high morbidity and mortality in the US and worldwide [1]. 

Arsenic trioxide is highly efficacious for APL and could be a promising drug against not only leukemia but also solid tumors, including PDAC [5,8,27]. Moreover, ATO treatment was found to inhibit the viability of PDAC stem cells in vitro and in vivo [28]. 

Heme oxygenases (HOs) are responsible for the degradation of heme to carbon monoxide (CO), ferrous iron, and biliverdin products. Biliverdin is then rapidly converted to bilirubin by biliverdin reductase [11]. Three mammalian HO isoforms have been identified, namely HO-1, HO-2, and HO-3. HO-1 is the inducible isoform [11]. HO-1 is overexpressed in PDAC tissues [15]. Several investigators, including our lab, have highlighted the critical role of HO-1 in different solid cancers’ development and progression, including PDAC [13,14,15,29]. Increased HO-1 promoted tumor progression and accelerated metastasis in a PDAC model [15]. Interestingly, multiple chemotherapy studies have suggested that increased cellular HO-1 levels in response to chemotherapy is associated with chemoresistance [14,15,30]. 

Drug resistance causes failure of ATO treatment in solid tumors, and the requirement for higher doses is associated with increased side effects, including cardiotoxicity [31]. Clearly, avoiding ATO toxicity and eliminating drug resistance are critically needed for the applications of ATO in solid tumors combined therapy. Therefore, new strategies are needed to enhance the antitumor activity of ATO and minimize its side effects.

Studies showed that ATO reduced tumor cell survival, induced apoptosis, and increased autophagosome accumulation in PDAC tumor cells [22]. The findings of our present study indicate that HO-1 inhibitors significantly potentiate the killing effects of ATO in PDAC cells. This combined therapy induced apoptosis, disrupted antioxidants, and increased ROS production, which facilitated the modulation of autophagy in PDAC cells.

The cytotoxicity of ATO in combination with HO-1 inhibition against human PDAC cells was assessed using several parameters. Different concentrations and times of exposure of single and combined therapy were tested in our experiments. Treatment with ATO induced HO-1 in multiple PDAC cell lines. Cell viability was significantly inhibited in a dose-dependent manner by ATO, which was further augmented by HO-1 inhibition.

Autophagy is an evolutionarily conserved physiological process mediated by intracellular lysosomes and plays an important role in maintaining the stability of the internal environment in response to various stresses, including oxidative stress [32]. Autophagy can be induced by metabolic stress, organelle dysfunction, or protein aggregation. It is involved in various aspects of cell biology including infections, immune function, and resistance to certain therapies.

In cancers, resistance to cancer treatments, including chemotherapy and immunotherapy, has been attributed to autophagy [33,34]. The importance of autophagy in PDAC has been reported in several publications [35,36]. Unlike normal pancreatic ducts, PDAC cell lines have shown constitutively activated autophagy, making them uniquely sensitive to autophagy inhibition [35]. It has been shown that inhibition of autophagy in nude mice led to reduced cell proliferation in vitro and tumor growth in vivo [35]. An impaired progression of pre-malignant to invasive PDAC was shown in animal experiments with ATG5 deletion in a PDAC mouse model using the *Kras* mutant and loss of a single *Trp53* allele [37]. Treatment of murine cell lines (independent of their *Trp53* status) with CQN or hydroxychloroquine led to decreased proliferation, increased DNA damage, and apoptosis. However, the role of autophagy in KRAS- and/or TP53-mediated carcinogenesis in PDAC is still not clear and needs to be investigated [37,38].

Different chemotherapy agents were shown to modulate autophagy in PDAC. This suggests that blocking autophagy could augment the therapeutic efficacy of these agents, and therefore, combining autophagy inhibition with the standard treatment approach may reduce chemoresistance in PDAC [39].

Based on our previous data showing that HO-1 inhibition preferentially targets antioxidant systems in PDAC cells, induces oxidative stress, and increases ROS production [14,30], we suggested that HO-1 inhibition is able to perturb autophagy in PDAC cells.

It has been shown that cancer cells, including PDAC cells, can function with higher levels of intracellular ROS than normal cells in culture and in vivo [40]. Under continuous intrinsic oxidative stress, cancer cells activate ROS-scavenging systems to adapt to stress by inducing multiple endogenous antioxidant compounds and enzymes including HO-1 [14,40]. However, increasing ROS levels to exceed cellular antioxidant capacity may become cytotoxic [40,41]. In our experiments, the levels of HO-1 were significantly increased under ATO treatment, and inhibiting HO-1 resulted in the elevation of ATO-induced ROS and a disrupted GSH system, indicating increased oxidative stress. Both ATO and HO-1 inhibitors increased the expression levels of the autophagy markers ATG5 and LC3B in PDAC cells. This increase was mediated by increased ROS production as NAC pretreatment significantly attenuated the protein induced by either treatment. We used CQN as a known inhibitor of autophagy. CQN and its derivatives are among the drugs that effectively inhibit autophagy by blocking lysosomal acidification and autophagosome degradation [26].

The association between apoptosis and autophagy has been described, wherein inhibiting autophagy leads to apoptosis-mediated cell death [42]. Gene silencing of autophagy proteins including Beclin1, ATG5, and others inhibited autophagy and sensitized drug-resistant cancer cells [43]. Moreover, blocking autophagy using pharmacological blockers (CQN), or silencing of ATG5 and ATG7, significantly enhanced the killing of imatinib-resistant cells in chronic myelogenous leukemia [43]. Similarly, the knockdown of ATG7 or Beclin1 or treatment with CQN sensitized cancer cells to epidermal growth factor receptor blocking antibody (Cetuximab) and increased apoptosis [44]. Finally, inhibiting autophagy using miRNA-29c increased gemcitabine-induced pancreatic cancer cell apoptosis [45].

Our results suggested that ATO and HO-1 inhibition induce apoptosis is via ROS-mediated autophagy inhibition. This mechanism is regulated by the p38 MAPK signaling pathways.

In summary, our results show that inhibiting the antioxidant role of HO-1 can modulate autophagy in PDAC cells. Since autophagy is an important mechanism of ATO resistance in PDAC cells, combined HO-1 inhibition with ATO will provide a basis for future preclinical trials as a combinatory therapeutic approach for PDAC.

## Figures and Tables

**Figure 1 biomedicines-11-02580-f001:**
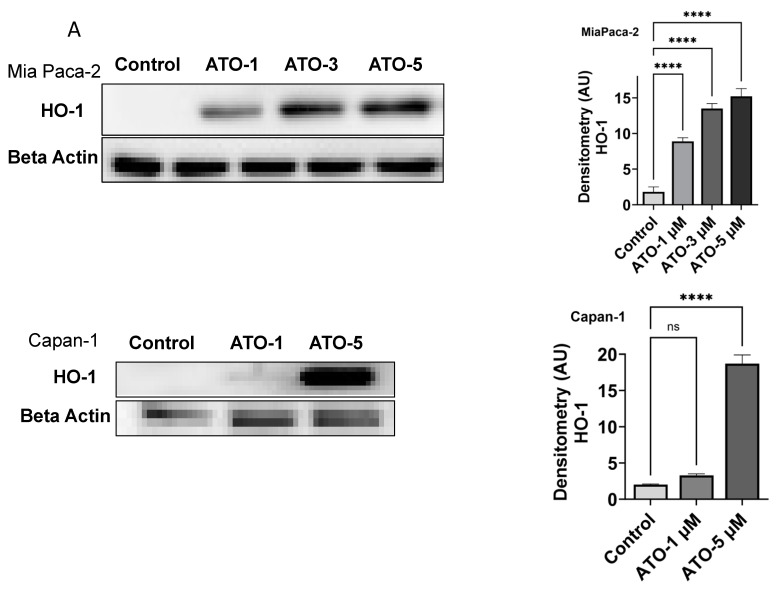
Induction of HO-1 in human PDAC cells upon treatment with ATO. (**A**) The effect of ATO on HO-1 expression was examined through Western blot analysis in PDAC cells MiaPaca-1 and Capan-1. Cells were treated with different concentrations of ATO for 24 h. Protein was collected and run for immunoblotting as described in the Section 2. Densitometric analysis is shown next to each blot (*n* = 3, *p* < 0.05). (**B**) Increased HO-1 expression after ATO treatment in Colo-357 shown by immunoblotting. Confocal microscopy confirms ATO induction of HO-1 in Colo-357. (**C**) HO-1 upregulation is via activation of the p38 MAPK pathway in response to ATO. MiaPaca-2 and Capan-1 cells were treated with indicated concentrations of ATO to follow pp38 MAPK and HO-1 expression using Western blot analysis. Data are representative of three independent experiments. (** *p* < 0.01, **** *p* < 0.0001. *n* = 3).

**Figure 2 biomedicines-11-02580-f002:**
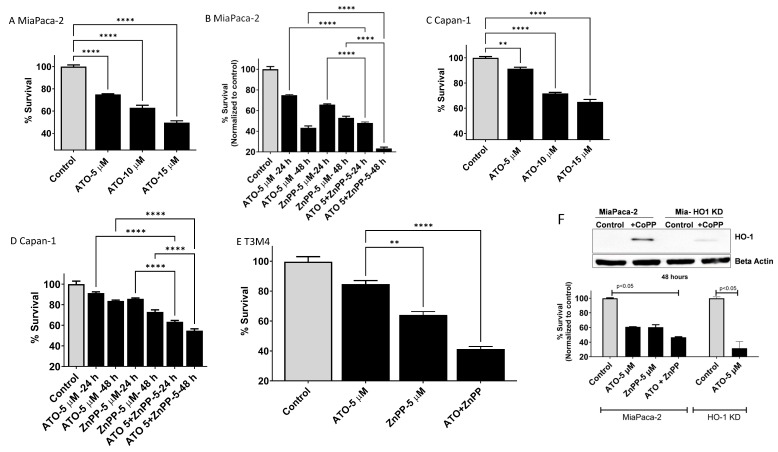
Time- and dose-dependent ATO effects on PDAC cells’ survival. Cell survival and growth inhibition was measured using the MTT assay. (**A**) MiaPaca-2 pancreatic cell line was treated with indicated concentration of ATO (5–15 μM) for 24 h. (**B**) MiaPaca-2 cells’ survival after 24 and 48 h culture in combination with HO-1 inhibition. (**C**) Capan-1 response to ATO after 24 h incubation and (**D**) combining ATO with HO-1 inhibition for 24 and 48 h. (**E**) T3M4 PDAC cells’ response to ATO and HO-1 inhibition after 24 h. (**F**) Survival of parent and HO-1 KO MaiaPaca-2 cells in response to ATO as compared to own controls. Insert showing HO-1 KO in response to CoPP for 24 h as compared to parent cells. (** *p* < 0.01, **** *p* < 0.0001. *n* = 3).

**Figure 3 biomedicines-11-02580-f003:**
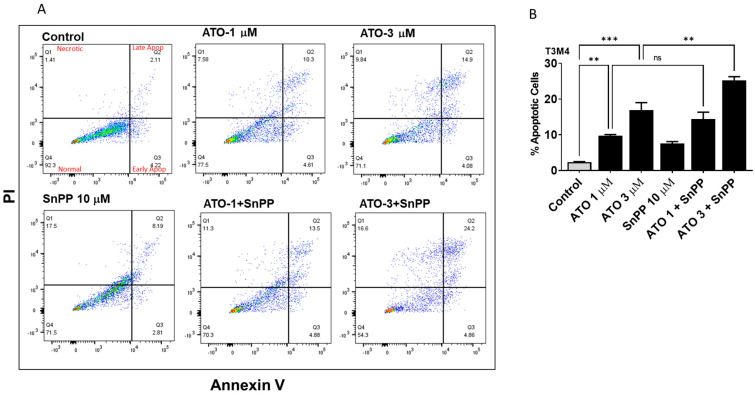
ATO and HO-1 inhibition promoted apoptosis in PDAC cells. PDAC cells were treated with vehicle, HO-1 inhibitor, and/or ATO. Apoptotic cells were counted using flow cytometry. (**A**) Example of flow results for T3M4 PDAC cells treated with either ATO, HO-1 inhibitor (SnPP), or combination. (**B**) Densitometric analysis paragraphs of total apoptotic cells of T3M4. (**C**) Apoptosis in T3M4 cells treated with the HO-1 inhibitor (ZnPP) and ATO. (**D**) Statistical paragraphs of T3M4 apoptosis in (**C**). (**E**) Apoptosis in MiaPaca-2 with the HO-1 inhibitor ZnPP and ATO. (**F**) Statistical paragraphs of MiaPaca-2 apoptosis in (E) (* *p* < 0.05, ** *p* < 0.01, *** *p* < 0.001, *n* = 3).

**Figure 4 biomedicines-11-02580-f004:**
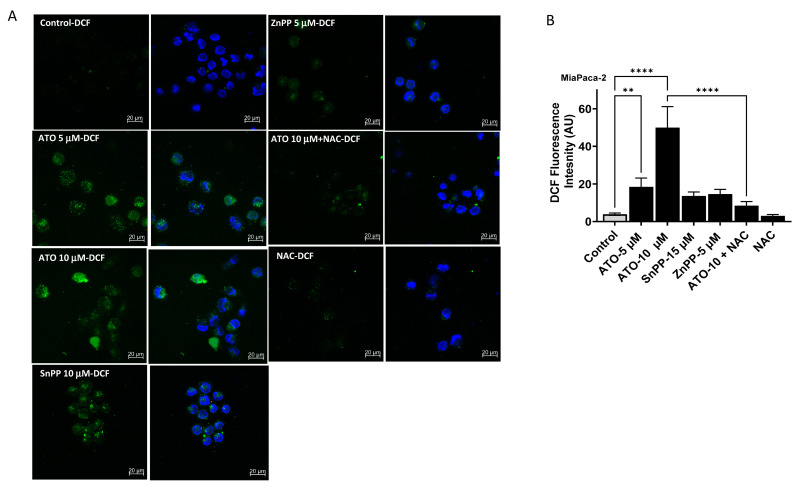
Effects of ATO, HO-1 inhibition, and combination on the levels of intracellular ROS as measured through confocal microscopy using an oxidation-sensitive fluorescent probe, 2′,7′-dichlorodihydrofluorescein diacetate (DCFH-DA), and Dihydroethidium (DHE). (**A**) ATO increased DCF fluorescence in MaiPaca-2 cells when treated as described in the Section 2. (**B**) Densitometric analysis of results in (**A**). (**C**) Increased DHE fluorescence in MiaPaca-2 cells after ATO treatment. (**D**) Oxidized glutathione (GSSG) in MiaPaca-2 and Capan-1 cells treated with ATO, HO-1 inhibitor, or combination. (**E**) Effects of NAC on ATO-induced antiproliferative effects in Colo-357, CD18/HPAF, and MiaPac-2 cells (* *p* < 0.05, ** *p* < 0.01, *** *p* < 0.001, **** *p* < 0.0001, *n* = 3). Scale bar: 20 μm.

**Figure 5 biomedicines-11-02580-f005:**
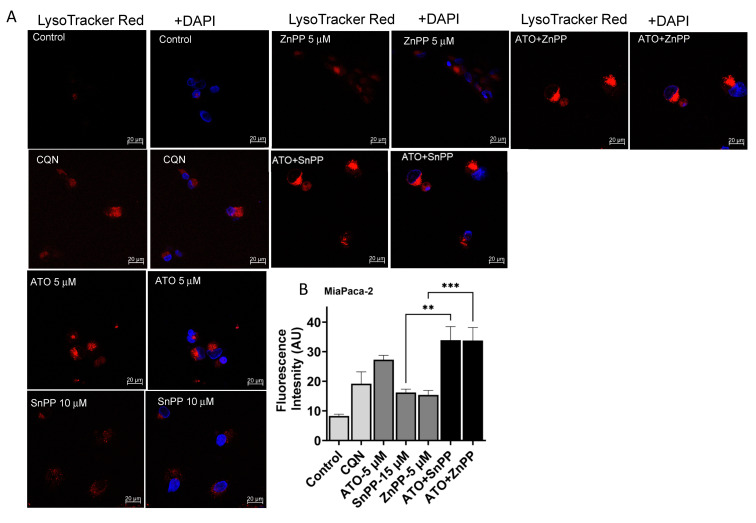
ATO and HO-1 inhibition disrupts lysosomal accumulation. Representative fluorescence images of LysoTracker Red staining of MiaPaca-2 cells for control and ATO- and CQN-treated cells (**A**) Both ATO and inhibiting HO-1 increased the fluorescence intensity of LysoTracker Red. (**B**) LysoTracker densitometry results show significantly increased lysosomal levels for cells treated with HO-1 inhibitors or CQN in comparison to the untreated control (** *p* < 0.01, *** *p* < 0.001, *n* = 3). Scale bar: 20 μm.

**Figure 6 biomedicines-11-02580-f006:**
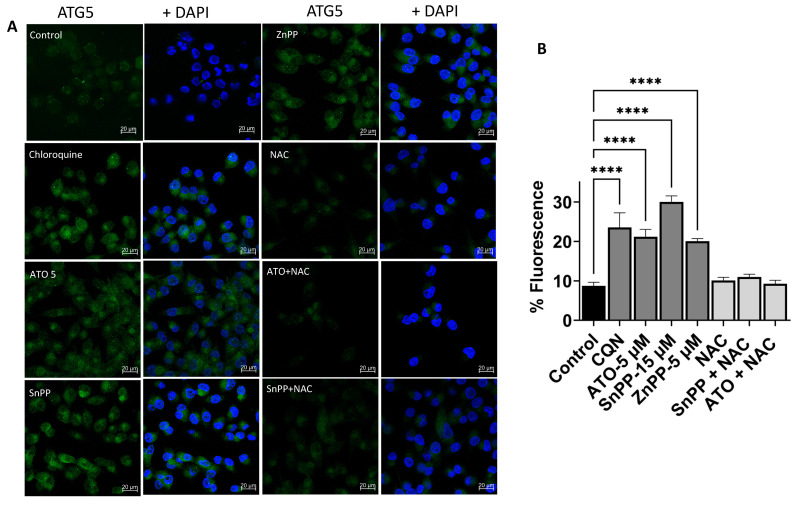
Inhibiting HO-1 suppresses autophagy. (**A**) Fluorescence photographs of MiaPaca-2 cells treated with ATO (5 μM), SnPP (10 μM), NAC (1 mM), chloroquine (15 μM), or combinations for 24 h. Nuclei were stained with DAPI (blue). Cells were treated with DMSO as a control. ATG5 was used as autophagy marker. (**B**) Average fluorescence of autophagosomes (ATG5 positive) per cell was calculated. Scale bar: 20 μm. (**C**) Immunoblotting of autophagy proteins for cells treated with ATO and/or HO-1 inhibitors showing increased LC3, ATg5, and p62 protein expression in response to ATO and HO-1 inhibition as compared to control. Densitometric analysis of each protein is shown as compared to control (* *p* < 0.05, ** *p* < 0.01, **** *p* < 0.0001, *n* = 3). Scale bar: 20 μm.

## Data Availability

The data presented in this study are available in this article.

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
