# Peer review of "Heme Oxygenase-1 Inhibition Modulates Autophagy and Augments Arsenic Trioxide Cytotoxicity in Pancreatic Cancer Cells"

_biomedicines, 2023, doi:10.3390/biomedicines11092580_

Round 1
Reviewer 1 Report
1. The authors need to mention the media used for cell culture and the source of CD18/HPAF cells in the material section.
2. The authors need to declare the source of the HO-1 inhibitors, SnPP and ZnPP in the material section.
3. The authors have to explain their inconsistant use of cell lines. throughout the manuscript:
- They used MIA-PaCa2 and Capan1 cells to show ATO increased HO-1 expression in Fig.1A. - Fig1B shows confocal data for Colo357 and Western blot data. Why did the authors use Colo357 cells for the confocal experiment and why didn't they include confocal data for MIA-PaCa2 and Capan1?
- In Fig.1C, the authors mix Western blot data showing an ATO-induced pP38 increase in MIA-PaCa2 and Capan1 and the effect of P38 inhibitors in another cell line, T3M4. Why did they swich cell lines? The authors need to show the effect of the p38 inhibitors in MIA-PaCa2 and Capan1 cells.
- Which cells were used in Fig. 4C? Please indicate in the legend.
- Why did the authors use Colo357 and yet another cell line CD18/HPAF to show the effect of NAC on ATO-induced cytotoxcity instead of Capan1 or T3M4 cells in Fig.4D?
4. The authors need to re-organize their figures:
- Subpanels should be clearly labeled A,B,C etc (Fig2-6)
- Subpanels should include the name of the cell line so that the reader does not have to constantly look at the figure legends
5. Inconsistant use of the HO-1 inhbitors, SnPP and ZnPP:
- The authors use SnPP for treatment of T3M4 cells (Fig.3 A,B) and ZnPP for treatment of MIA-PaCa2 cells (Fig.3C, D) Why did the authors use different inhibitors for different cell lines?
6. The lanes of the Western blot in Fig.6 need to be clearly labeled.
7. The authors added asterisks to the band intensity graphs in Fig.6, but failed to indicate which groups they are comparing.
8. The manuscript lacks in vivo data that could potentially show that a lower ATO dose in combination with HO-1 inhibitors has indeed better anticancer actvity AND less toxic side effects than a higher ATO dose alone. Adding in vivo data would greatly increase the significance of the findings.
Spelling errors: It should read MIA PaCa instead of "Mai-Paca"(lines 179, 201, 250)
Reviewer 2 Report
Authors have submitted quite a fascinating manuscript. The experimental design and the results seem to be exquisite and valid.
However, some points indicated in the attachment file would be considered to improve this article.

Author Response
Reviewer response attached.

Reviewer 3 Report
In this manuscript Ahmad et al., shows the effect of arsenic trioxide (ATO), inhibitors of heme oxygenase-1 (HO-1) and a ROS scavenger, in cell proliferation, cell death and autophagy induction assays. Although the work seems complete when approached from different points, there are several methodological deficiencies and in the presentation of results that detract from the quality of the work. They are listed below:
Major comments:
The expression level reflected in the densitometry of Figure 1A is different from that seen in the representative blot. For example, the western blot of the figure corresponding to Mia Paca-2, ATO 5, shows an expression of ~1 with respect to b-actin, but the densitometry shows ~180; similar with other samples of both cell lines The same occurs in the experiments of Figures 1C and 6c. Explain how the data was normalized.
Line 183, figure 2C: the graph shows decreased sensitivity but not insensitivity to ATO. What are they based on to say that they are insensitive?
Figure 2b,d: Is there a statistically significant difference when comparing ZnPP vs ATO+ZnPP? to really say that the combination is effective, and that it is not the effect of the inhibitor alone
Figure 3a: it is requested to show the dot plot of the autofluorescence
Both the title and the results in Figure 6 mention a disruption of autophagy with HO-1 inhibitors, which result or observation allows them to assert that "autophagy is being disrupted"?
Minor comments:
Line 20: GSH levels are not actually measured, but GSSG levels.
Line 166-167: actually the densitometric analysis is above the blot
Unify the location and style of the letters that indicate figure 1A or 2a; in figure 1 they are capital letters in the upper left corner, in figure 2 the letter is lower case, between parentheses and at the bottom of each graph. The same happens in the remaining figures
Figure 2f: Can the normalization of the effect of HO1-KD be shown with respect to its own control? considering that the control presents a decreased viability (~70%), this in order to compare the effect of the pharmacological inhibitor of HO-1 ZnPP vs the silencing of HO-1
Line 220: statistical paragraphs is correct?
Line 235: Homogenize N-acetylcysteine, in abstract it says N-acetyl cysteine.
Line 310, figure 6c: It is requested to expand this description and in the image of the western blot to indicate which sample is in each line
Round 2
Reviewer 1 Report
The authors have adequately addressed all comments.
Author Response
We thank the reviewer for his comment
Reviewer 3 Report
Thanks to the authors for attending to the comments, I consider that the work is solid enough to be published
Author Response
we thank the reviewer for his comment